# Sequential Memory with Temporal Predictive Coding

**Mufeng Tang, Helen Barron, Rafal Bogacz**
MRC Brain Network Dynamics Unit, University of Oxford, UK
{mufeng.tang, helen.barron, rafal.bogacz}@bndu.ox.ac.uk

## Abstract

Forming accurate memory of sequential stimuli is a fundamental function of biological agents. However, the computational mechanism underlying sequential memory in the brain remains unclear. Inspired by neuroscience theories and recent successes in applying predictive coding (PC) to *static* memory tasks, in this work we propose a novel PC-based model for *sequential* memory, called *temporal predictive coding* (tPC). We show that our tPC models can memorize and retrieve sequential inputs accurately with a biologically plausible neural implementation. Importantly, our analytical study reveals that tPC can be viewed as a classical Asymmetric Hopfield Network (AHN) with an implicit statistical whitening process, which leads to more stable performance in sequential memory tasks of structured inputs. Moreover, we find that tPC exhibits properties consistent with behavioral observations and theories in neuroscience, thereby strengthening its biological relevance. Our work establishes a possible computational mechanism underlying sequential memory in the brain that can also be theoretically interpreted using existing memory model frameworks.

## 1 Introduction

The ability to memorize and recall sequences of events with temporal dependencies is crucial for biological memory systems that also underpins many other neural processes [1–3]. For example, forming correct memories of words requires humans to memorize not only individual letters but also the sequential order following which the letters appear (e.g., "cat" and "act"). However, despite extensive research into models of *static*, temporally unrelated memories from both neuroscience and machine learning [4–10], computational modeling of *sequential* memory is not as developed. Existing models for sequential memory are either analytically intractable or have not yet been systematically evaluated in challenging sequential memory tasks [11–15], which hinders a comprehensive understanding of the computational mechanism underlying sequential memory, arguably a more general form of memory in the natural world than static memories.

In this work, we propose a novel approach to modeling sequential memory based on predictive coding (PC) [16–18], a biologically plausible neural network model able to reproduce many phenomena observed in the brain [19], which has also shown to have a close relationship to backpropagation in artificial neural networks [20, 21]. Using PC to model sequential memory is motivated by two key factors: Firstly, neuroscience experiments and theories have suggested that (temporal) predictive processing and memory are two highly related computations in the hippocampus, the brain region crucial for memory [22, 23]. Secondly, the modeling of static memories (e.g., a single image) using PC has recently demonstrated significant success [8, 10, 24, 25], raising the question of whether PC can also be employed to model sequential memory. To take into account the temporal dimension in sequential memory, in this work we adopt a temporal extension of the original PC models, which has been employed in filtering problems and in modeling the visual system [26–29]. Here we investigate its performance in sequential memory tasks. Our contributions can be summarized as follows:

37th Conference on Neural Information Processing Systems (NeurIPS 2023).

- We propose *temporal predictive coding* (tPC), a family of PC models capable of sequential memory tasks that inherit the biologically plausible neural implementation of classical PC models [18];

- We present an analytical result showing that the single-layer tPC can be viewed as the classical Asymmetric Hopfield Network (AHN) performing an implicit *statistical whitening* step during memory recall, providing a possible mechanism of statistical whitening in the brain [30–32];

- Experimentally, we show that the whitening step in single-layer tPC models results in more stable performance than the AHN and its modern variants [15] in sequential memory, due to the highly variable and correlated structure of natural sequential inputs;

- We show that tPC can successfully reproduce several behavioral observations in humans, including the impact of sequence length in word memories and the primacy/recency effect;

- Beyond memory, we show that our tPC model can also develop context-dependent representations [2, 11] and generalize learned dynamics to unseen sequences, suggesting a potential connection to cognitive maps in the brain [33, 34].

## 2 Background and related work

**Predictive Coding Models for Static Memory**   The original PC model for memory follows a hierarchical and generative structure, where higher layers generate top-down predictions of lower layers' activities and the network's learning and inference are driven by the minimization of all the prediction errors [8]. Since the memorized patterns minimize the prediction errors in PC, they can be considered attractors of the energy function defined as the (summed) prediction errors, which is similar to the energy perspective of Hopfield Networks (HNs) [4]. Subsequent research has also shown that PC models for memory can be formulated as recurrent networks to account for the recurrently connected hippocampal network [10, 25], and that continual memorization and recall of a stream of patterns can be achieved by combining the hierarchical PC model with conjugate Bayesian updates [24]. Despite these abundant investigations into the memory capability of PC models, they all focused on static memories. Although the BayesPCN model [24] is capable of recalling patterns memorized in an online manner, the memories stored in this model are still static, order-invariant patterns, rather than sequences of patterns with underlying temporal dependencies.

**Predictive Coding with Time**   PC was extended to include temporal predictions by earlier works (see [35] for a review) noticing its relationship to Kalman filters [36]. However, the Kalman filtering approach to temporal predictions sacrifices the plausible neural implementation of PC models due to non-local computations, and a recent work proposed an alternative way of formulating PC with temporal predictions that inherits the plausible implementation of static PC, while approximating Kalman filters [37]. However, none of these models were examined in memory tasks. Broadly, models based on temporal predictions were proposed in both neuroscience and machine learning [29, 38–40]. However, these models are either trained by the implausible backpropagation or rely on complex architectures to achieve temporal predictions. In this work, our tPC model for sequential memory is based on the model in [37], which inherits the simple and biologically plausible neural implementation of classical PC models that requires only local computations and Hebbian plasticity.

**Hopfield Networks for Sequential Memory**   Although there exist other models of sequential memory [11–13], these models are mostly on the conceptual level, used to provide theoretical accounts for physiological observations from the brain, and are thus hard to analyze mathematically. Therefore, here we focus our discussion on the AHN [14] and its modern variants [15], which extended the HN [4] to account for sequential memory, and have a more explicit mathematical formulation. Denoting a sequence of $P + 1$ patterns $\mathbf{x}^\mu$ ($\mu = 1, ..., P + 1$), $\mathbf{x}^\mu \in \{-1, 1\}^N$, the $N \times N$ weight matrix of an AHN is set as follows:

$$\mathbf{W}^{AHN} = \sum_{\mu=1}^P \mathbf{x}^{\mu+1}(\mathbf{x}^\mu)^\top \tag{1}$$

Notice that it differs from the static HN only by encoding the asymmetric autocovariance rather than the symmetric covariance $\sum_{\mu=1}^P \mathbf{x}^\mu(\mathbf{x}^\mu)^\top$, thus the name. The single-shot retrieval, which we define as $R$, is then triggered by a query, $\mathbf{q} \in \{-1, 1\}^N$:

$$R^{AHN}(\mathbf{q}) = \text{sgn}(\mathbf{W}^{AHN}\mathbf{q}) = \text{sgn}\left(\sum_{\mu=1}^P \mathbf{x}^{\mu+1}(\mathbf{x}^\mu)^\top \mathbf{q}\right) \tag{2}$$

The retrieval process in Eq. 2 can be viewed as follows: the query $\mathbf{q}$ is first compared with each $\mathbf{x}^\mu$ using a dot product function $(\mathbf{x}^\mu)^\top \mathbf{q}$ that outputs a similarity score, then the retrieval is a weighted sum of all $\mathbf{x}^{\mu+1}$ ($\mu = 1, ..., P$) using these scores as weights. Thus, if $\mathbf{q}$ is identical to a certain $\mathbf{x}^\mu$, the next pattern $\mathbf{x}^{\mu+1}$ will be given the largest weight in the output retrieval. Following the Universal Hopfield Network (UHN) framework [7], we can generalize this process to define a retrieval function of a general sequential memory model with real-valued patterns $\mathbf{x}^\mu \in \mathbb{R}^N$:

$$R^{UHN}(\mathbf{q}) = \sum_{\mu=1}^P \mathbf{x}^{\mu+1} \operatorname{sep}\left(\operatorname{sim}(\mathbf{x}^\mu, \mathbf{q})\right) \tag{3}$$

where sim is a similarity function such as dot product or cosine similarity, and sep is a separation function that separates the similarity scores i.e., emphasize large scores and de-emphasize smaller ones [7]. When sim is dot product and sep is an identity function, we get the retrieval function of the original AHN (for binary patterns an ad-hoc sgn function can be applied to the retrievals). Chaudhry et al. [15] have shown that it is possible to extend AHN to a model with polynomial sep function with a degree $d$, and a model with softmax sep function, which we call "Modern Continuous AHN" (MCAHN):

$$R^{AHN}(\mathbf{q}, d) = \sum_{\mu=1}^P \mathbf{x}^{\mu+1} \left((\mathbf{x}^\mu)^\top \mathbf{q}\right)^d \tag{4}$$

$$R^{MCAHN}(\mathbf{q}, \beta) = \sum_{\mu=1}^P \mathbf{x}^{\mu+1} \operatorname{softmax}\left(\beta(\mathbf{x}^\mu)^\top \mathbf{q}\right) \tag{5}$$

where $\beta$ is the temperature parameter that controls the separation strength of the MCAHN. Note that these two models can be respectively viewed as sequential versions of the Modern Hopfield Network [5] and the Modern Continuous Hopfield Network [6], which is closely related to the self-attention mechanism in Transformers [41]. However, the family of AHNs has not yet been investigated in sequential memory tasks with structured and complex inputs such as natural movies.

**Other Models for Sequential Memory** Beyond Hopfield Networks, many other computational models have been proposed to study the mechanism underlying sequential memory. Theoretical properties of self-organizing networks in sequential memory were discussed as early as in [42]. In theoretical neuroscience, models by Jensen et al. [43] and Mehta et al. [44] suggested that the hippocampus performs sequential memory via neuron firing chains. Other models have suggested the role of contextual representation in sequential memory [11, 45], with contextual representations successfully reproducing the recency and contiguity effects in free recall [46]. Furthermore, Howard et al. [47] proposed that sequential memory is represented in the brain via approximating the inverse Laplacian transform of the current sensory input. However, these models were still at the conceptual level, lacking neural implementations of the computations. Recurrent networks with backpropagation and large spiking neural networks also demonstrate sequential memory [48, 49]. We compare our model with [48] to validate tPC's alignment with behavior.

Our model is also closely related to the concept of cognitive map in the hippocampal formation [50–52], which is often discussed within the context of sequence learning to explain knowledge abstraction and generalization. In this work, we present two preliminary results related to cognitive maps, showing that our tPC model can 1) disambiguate aliased observation via latent representations and 2) generalize with simple sequential dynamics as a result of performing sequential memory [52]. However, as this work centers on memory, we leave cognitive maps for future explorations of tPC.

## 3 Models

In this section, we introduce the tPC models by describing their computations during **memorization** and **recall** respectively, as well as the **neural implementations** of these computations. We describe the single-layer tPC first, and then move to the 2-layer tPC.

### 3.1 Single-layer tPC

**Memorization** The most straightforward intuition behind "good" sequential memory models is that they should learn the transition between every pair of consecutive patterns in the sequence, so that accurate recall of the full sequence can be achieved recursively by recalling the next pattern based on the current one. Given a sequence of real-valued patterns $\mathbf{x}^\mu$, $\mu = 1, ..., P + 1$, this intuition can be formalized as the model minimizing the following loss at each time-step:

$$\mathcal{F}_\mu(\mathbf{W}) = \|\mathbf{x}^\mu - \mathbf{W}f(\mathbf{x}^{\mu-1})\|_2^2 \tag{6}$$

which is simply the squared temporal prediction error. $\mathbf{W}$ is the weight parameter of the model, and $f(\cdot)$ is a nonlinear function. Similar to static PC models [8, 10], we assume that the model has two populations of neurons: value neurons that are loaded with the inputs $\mathbf{x}^\mu$ at step $\mu$, and error neurons representing the temporal prediction error $\boldsymbol{\varepsilon}^\mu := \mathbf{x}^\mu - \mathbf{W}f(\mathbf{x}^{\mu-1})$. To memorize the sequence, the weight parameter $\mathbf{W}$ is updated at each step following gradient descent:

$$\Delta\mathbf{W} \propto -\partial\mathcal{F}_\mu(\mathbf{W})/\partial\mathbf{W} = \boldsymbol{\varepsilon}^\mu f(\mathbf{x}^{\mu-1})^\top \tag{7}$$

and the model can be presented with the sequence for multiple epochs until $\mathbf{W}$ converges. Note that the model only has one weight parameter for the whole sequence, rather than $P$ weight parameters for a sequence of length $P+1$.

**Recall**    During recall, the weight matrix $\mathbf{W}$ is fixed to the learned values, and the value neurons no longer receive the correct patterns $\mathbf{x}^\mu$. Instead, while trying to recall the pattern $\mathbf{x}^\mu$ based on the query $\mathbf{q}$, the value neurons are updated to minimize the squared temporal prediction error based on the query $\mathbf{q}$ and the learned $\mathbf{W}$:

$$\mathcal{F}_\mu(\hat{\mathbf{x}}^\mu) = \|\hat{\mathbf{x}}^\mu - \mathbf{W}f(\mathbf{q})\|_2^2 \tag{8}$$

where we denote the value neurons' activities during recall as $\hat{\mathbf{x}}^\mu$ to differentiate it from the memorized patterns $\mathbf{x}^\mu$. The value neurons then perform the following inferential dynamics to minimize the loss $\mathcal{F}_\mu(\hat{\mathbf{x}}^\mu)$:

$$\dot{\hat{\mathbf{x}}}^\mu \propto -\partial\mathcal{F}_\mu(\hat{\mathbf{x}}^\mu)/\partial\hat{\mathbf{x}}^\mu = -\boldsymbol{\varepsilon}^\mu \tag{9}$$

and the converged $\hat{\mathbf{x}}^\mu$ is the final retrieval. Note that the error neurons' activities during recall are defined as $\boldsymbol{\varepsilon}^\mu := \hat{\mathbf{x}}^\mu - \mathbf{W}f(\mathbf{q})$, which is also different from their activities during memorization.

In the case of sequential memory, there are two types of recall. We define the first type as "online" recall, where the query $\mathbf{q}$ at each step $\mu$ is the ground-truth pattern at the previous step $\mathbf{x}^{\mu-1}$. It is called online as these ground-truth queries can be viewed as real-time online feedback during the sequence recall. In this case, the original $\mathbf{x}^\mu$ will define a memory attractor as it defines an optimum of the loss, or energy in Eqs. 6 and 8. The second type is referred to as "offline" recall, where $\mathbf{q}$ is the recall from the previous step i.e., $\hat{\mathbf{x}}^{\mu-1}$, except at the first step, where a ground-truth $\mathbf{x}^1$ is supplied to elicit recall of the whole sequence. This is called offline as there is no real-time feedback. In this case, errors from earlier steps may accumulate through time and $\mathbf{x}^\mu$ is no longer an ascertained attractor unless $\hat{\mathbf{x}}^{\mu-1} = \mathbf{x}^{\mu-1}$, which makes it more challenging and analogous to the replay of memories during sleep [53].

**Neural implementation**    A possible neural network implementation of these computations is shown in Fig. 1A, which is similar to that of static PC models [18] characterized by separate populations of value and error neurons. The difference from static models is that the predictions are now from the previous time-step $\mu - 1$. To achieve this, we assume that the value neurons are connected to the error neurons via two pathways: the direct pathway (the straight arrows between value and error neurons) and the indirect pathway through an additional population of inhibitory interneurons, which provides inhibitory inputs to the error neurons via $\mathbf{W}$. These interneurons naturally introduce a synaptic delay of one time-step, such that when the inputs from step $\mu - 1$ reach the error neurons through the indirect pathway, the error neurons are already receiving inputs from step $\mu$ via the direct pathway, resulting in the temporal error. Moreover, we assume that memory recall is a much faster process than the time-steps $\mu$ so that the interneurons can hold a short working memory of $\mathbf{q}$ during the (iterative) inferential dynamics in Eq. 9 at step $\mu$, which can be achieved by the mechanisms described in [54]. Notice that in this implementation, the learning rule (Eq. 7) is Hebbian and the inference rule (Eq. 9) is also local, inheriting the plausibility of static PC implementations [18].

## 3.2   2-layer tPC

Similar to multi-layer static PC models for memory, we can have multiple layers in tPC to model the hierarchical processing of raw sensory inputs by the neocortex, before they enter the memory system [8, 10]. We focus on a 2-layer tPC model in this work. In this model, we assume a set of *hidden* value neurons $\mathbf{z}^\mu$ to model the brain's internal neural responses to the sequential sensory inputs $\mathbf{x}^\mu$. The hidden neurons make not only hierarchical, top-down predictions of the current activities in the sensory layer like in static PC models [8], but also temporal predictions like in the single-layer tPC. We also assume that the sensory layer $\mathbf{x}^\mu$ does not make any temporal predictions in this case. Thus, this 2-layer tPC can be viewed as an instantiation of the hidden Markov model [55].

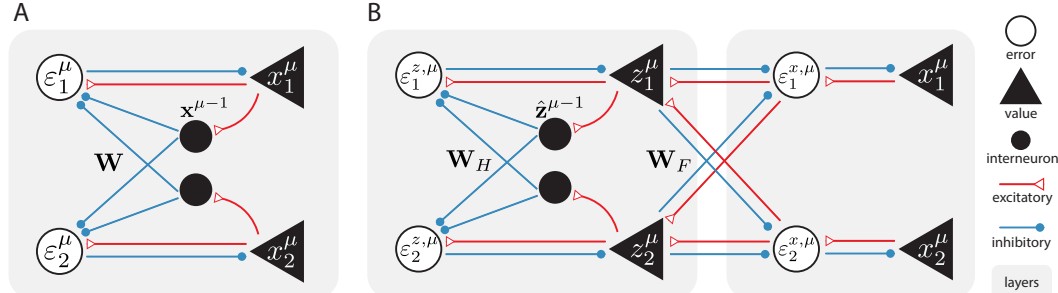

Figure 1: Neural implementations of the tPC models. A: single-layer tPC. B: 2-layer tPC.

**Memorization** During memorization, the 2-layer tPC tries to minimize the sum of squared errors at step $\mu$, with respect to the model parameters and the hidden activities:

$$\mathcal{F}_\mu(\mathbf{z}^\mu, \mathbf{W}_H, \mathbf{W}_F) = \|\mathbf{z}^\mu - \mathbf{W}_H f(\hat{\mathbf{z}}^{\mu-1})\|_2^2 + \|\mathbf{x}^\mu - \mathbf{W}_F f(\mathbf{z}^\mu)\|_2^2 \qquad (10)$$

where $\mathbf{W}_H$ governs the temporal prediction in the hidden state, $\mathbf{W}_F$ is the forward weight for the top-down predictions, and $\hat{\mathbf{z}}^{\mu-1}$ is the hidden state inferred at the previous time-step. During memorization, the 2-layer tPC follows a similar optimization processing to that of static hierarchical PC [8]. It first infers the hidden representation of the current sensory input $\mathbf{x}^\mu$ by:

$$\dot{\mathbf{z}}^\mu \propto -\partial \mathcal{F}_\mu(\mathbf{z}^\mu, \mathbf{W}_H, \mathbf{W}_F)/\partial \mathbf{z}^\mu = -\boldsymbol{\varepsilon}^{\mathbf{z},\mu} + f'(\mathbf{z}^\mu) \odot \mathbf{W}_F^\top \boldsymbol{\varepsilon}^{\mathbf{x},\mu} \qquad (11)$$

where $\odot$ denotes the element-wise product between two vectors, and $\boldsymbol{\varepsilon}^{\mathbf{z},\mu}$ and $\boldsymbol{\varepsilon}^{\mathbf{x},\mu}$ are defined as the hidden temporal prediction error $\mathbf{z}^\mu - \mathbf{W}_H f(\hat{\mathbf{z}}^{\mu-1})$ and the top-down error $\mathbf{x}^\mu - \mathbf{W}_F f(\mathbf{z}^\mu)$ respectively. After $\mathbf{z}^\mu$ converges, $\mathbf{W}_H$ and $\mathbf{W}_F$ are updated following gradient descent on $\mathcal{F}_\mu$:

$$\begin{aligned}
\Delta \mathbf{W}_H &\propto -\partial \mathcal{F}_\mu(\mathbf{z}^\mu, \mathbf{W}_H, \mathbf{W}_F)/\partial \mathbf{W}_H = \boldsymbol{\varepsilon}^{\mathbf{z},\mu} f(\hat{\mathbf{z}}^{\mu-1})^\top; \\
\Delta \mathbf{W}_F &\propto -\partial \mathcal{F}_\mu(\mathbf{z}^\mu, \mathbf{W}_H, \mathbf{W}_F)/\partial \mathbf{W}_F = \boldsymbol{\varepsilon}^{\mathbf{x},\mu} f(\mathbf{z}^\mu)^\top
\end{aligned} \qquad (12)$$

which are performed once for every presentation of the full sequence. The converged $\mathbf{z}^\mu$ is then used as $\hat{\mathbf{z}}^\mu$ for the memorization at time-step $\mu + 1$.

**Recall** After learning/memorization, $\mathbf{W}_H$ and $\mathbf{W}_F$ are fixed. We also assume that the hidden activities $\mathbf{z}^\mu$ are unable to store memories, by resetting their values to randomly initialized ones. Thus, the sequential memories can only be recalled through the weights $\mathbf{W}_H$ and $\mathbf{W}_F$. Again, the sensory layer has no access to the correct patterns during recall and thus needs to dynamically change its value to retrieve the memories. The loss thus becomes:

$$\mathcal{F}_\mu(\mathbf{z}^\mu, \hat{\mathbf{x}}^\mu) = \|\mathbf{z}^\mu - \mathbf{W}_H f(\hat{\mathbf{z}}^{\mu-1})\|_2^2 + \|\hat{\mathbf{x}}^\mu - \mathbf{W}_F f(\mathbf{z}^\mu)\|_2^2 \qquad (13)$$

where $\hat{\mathbf{x}}^\mu$ denotes the activities of value neurons in the sensory layer during recall. Both the hidden and sensory value neurons are updated to minimize the loss. The hidden neurons will follow similar dynamics specified in Eq. 11, with the top-down error $\boldsymbol{\varepsilon}^{\mathbf{x},\mu}$ defined as $\hat{\mathbf{x}}^\mu - \mathbf{W}_F f(\mathbf{z}^\mu)$, whereas the sensory neurons are updated according to:

$$\dot{\hat{\mathbf{x}}}^\mu \propto -\partial \mathcal{F}_\mu(\mathbf{z}^\mu, \hat{\mathbf{x}}^\mu)/\partial \hat{\mathbf{x}}^\mu = -\boldsymbol{\varepsilon}^{\mathbf{x},\mu} \qquad (14)$$

and the converged $\hat{\mathbf{x}}^\mu$ is the final retrieval. Similar to the single-layer case, if the converged $\hat{\mathbf{z}}^\mu$ is used for the recall at the next step directly, the recall is offline; on the other hand, if we query the model with $\mathbf{q} = \mathbf{x}^\mu$ i.e., the ground-truth and use the query to infer $\hat{\mathbf{z}}^\mu$, and then use $\hat{\mathbf{z}}^\mu$ for the recall at the next step, the recall is online.

**Neural implementation** A neural implementation of the computations above is shown in Fig. 1B. The hidden layer follows the same mechanism in the single-layer tPC, with interneurons introducing a synaptic delay for temporal errors and short-term memory to perform the inferential dynamics (Eq. 11). The connection between the hidden layer and the sensory layer $\mathbf{W}_F$ is modeled in the same way as in static PC models, which requires only Hebbian learning and local computations [18]. The memorization and recall pseudocode for the tPC models is provided in SM.

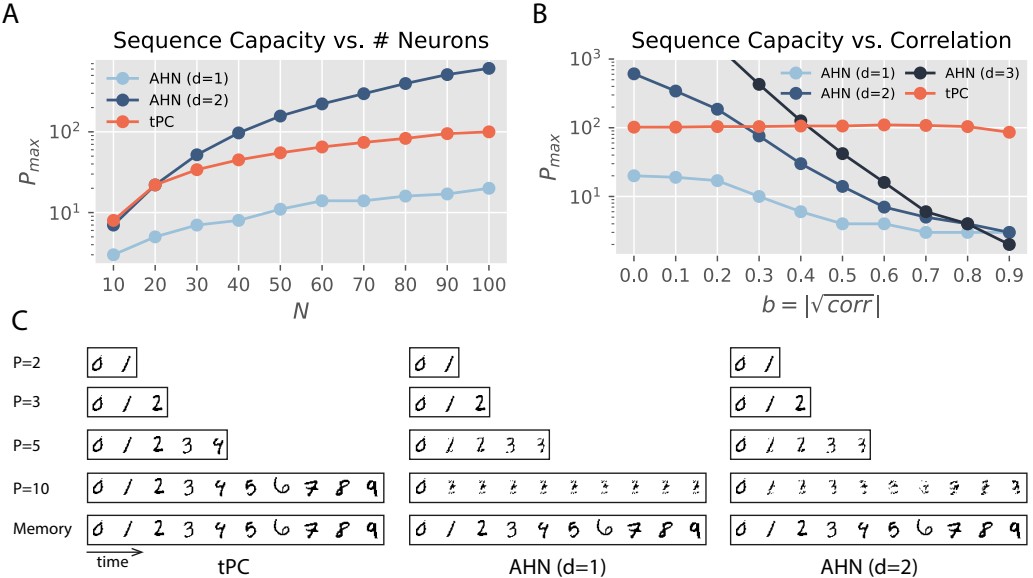

Figure 2: Comparison between single-layer tPC and AHNs. A: Capacity of models with uncorrelated binary patterns. B: Capacity of models with binary patterns with increasing feature correlations. C: Recall performance with sequences of binary MNIST digits.

## 4 Results

### 4.1 Theoretical relationship to AHNs

We first develop a theoretical understanding of single-layer tPC by relating it to AHNs:

**Property 1** *Assume, without loss of generality, a sequence of memories $\mathbf{x}^\mu \in \mathbb{R}^N$ ($\mu = 1, ..., P+1$) with zero mean. With an identity nonlinear function $f(x) = x$ in Eq. 6, the retrieval of the single-layer tPC with query $\mathbf{q}$, defined as $R^{tPC}(\mathbf{q})$, can be written as:*

$$R^{tPC}(\mathbf{q}) = \sum_{\mu=1}^{P} \mathbf{x}^{\mu+1}(\mathbf{M}\mathbf{x}^\mu)^\top \mathbf{M}\mathbf{q} \qquad (15)$$

*where $\mathbf{M}$ is an empirical whitening matrix such that:*

$$\langle \mathbf{M}\mathbf{x}^\mu (\mathbf{M}\mathbf{x}^\mu)^\top \rangle_\mu = \mathbf{I}_N \qquad (16)$$

*where $\langle \cdot \rangle_\mu$ is the expectation operation over $\mathbf{x}^\mu$'s.*

Proof of this property is provided in the SM. Essentially, this property implies that our single-layer tPC, in its linear form, can be regarded as a special case of the UHN for sequential memories (Eq. 3), with a "whitened dot product" similarity function where the two vectors $\mathbf{x}^\mu$ and $\mathbf{q}$ are first normalized and decorrelated (to have identity covariance $\mathbf{I}_N$) before the dot product. AHNs, on the other hand, calculate the dot product directly. Biologically, this property provides a possible mechanism of statistical whitening in the brain that, unlike earlier models of biological whitening with explicit objectives [30–32], performs this computation *implicitly* via the circuit shown in Fig. 1 that minimizes the temporal prediction errors.

### 4.2 Experimental comparison to AHNs

To understand how the whitening step affects the performance in sequential memory tasks, we compare our single-layer tPC with the family of AHNs exeperimentally. To ensure consistency to Property 1 above, we use an identity nonlinearity $f(x) = x$ for all these experiments. Empirically, we found that using a $tanh$ nonlinearity makes subtle differences irrelevant to our main discussion in this work, and we discuss it in SM.

**Polynomial AHNs** We first compare tPC with polynomial AHNs (Eq. 4) in sequences of uncorrelated binary patterns, where AHNs are known to work well [14, 15]. We plot their sequence capacity

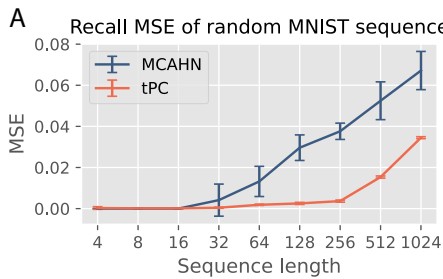 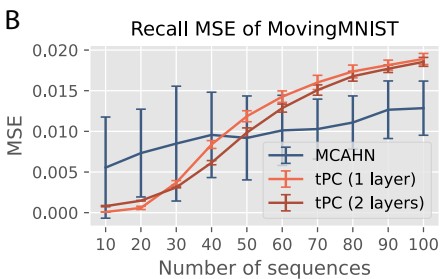

Figure 3: A: Recall MSE of MNIST sequences with increasing length; B: Recall MSE of MovingM-NIST sequences of a fixed length 10 but with an increasing number of sequences. Error bars obtained with 5 seeds

$P_{max}$ against the number of value neurons of the models i.e., the pattern dimension $N$. Here, $P_{max}$ is defined as the maximum length of a memorized sequence, for which the probability of incorrectly recalled bits is less than or equal to $0.01$. Fig. 2A shows that the capacity of our single-layer tPC is greater than that of the original AHN ($d = 1$) but smaller than that of a quadratic AHN ($d = 2$). Notice that the single-layer tPC has an identity sep function like the original AHN. Therefore the whitening operation has indeed improved the performance of the dot product sim function. Inspired by the decorrelation effect of statistical whitening, we then generated binary patterns with $N = 100$ *correlated* features, with a parameter $b$ controlling the level of correlation ($b = |\sqrt{\text{correlation}}|$). The approach that we followed to generate the correlated patterns is provided in SM. As shown in Fig. 2B, as the correlation increases, all AHNs up to $d = 3$ suffer from a quick decrease of capacity $P_{max}$, whereas the capacity of single-layer tPC almost remains constant. This observation is consistent with the theoretical property that the whitening transformation essentially decorrelates features such that patterns with any level of correlation are regarded as uncorrelated in tPC recall (Eq. 15). This result also explains the comparison in Fig. 2A: although the patterns generated in this panel are theoretically uncorrelated, the small correlation introduced due to experimental randomness will result in the performance gap between AHN and single-layer tPC.

We then investigate the performance of these models with sequences of binarized MNIST images [56] in Fig. 2C. It can be seen that the AHNs with $d = 1$ and $d = 2$ quickly fail as the sequence length $P$ reaches 3, whereas the single-layer tPC performs well. These results suggest that our tPC with a whitening step is superior to simple AHNs due to the inherently correlated structure of natural inputs such as handwritten digits. For all the experiments with binary patterns, we used the online recall mentioned above, and polynomial AHNs already fail in this simpler recall scenario.

**MCAHN**    Due to the quick failure of AHNs with a polynomial sep function, we now compare our single-layer tPC with the MCAHN (Eq. 5). In static memory tasks, it is known that the softmax separation leads to exponentially high capacity, especially when $\beta$ is high [7, 57]. In this work we use $\beta = 5$ for all MCAHNs. We first compare the performances of our single-layer tPC model and an MCAHN on random sequences of MNIST digits with varying lengths. Here we trigger recalls with online queries. Fig. 3A shows that the performance of our single-layer tPC, measured as the mean squared error (MSE) between the recalled sequence and the ground truth, is better than that of the MCAHN, further demonstrating the usefulness of the implicit whitening in our model.

Despite the superior performance of our model in this task, we note that random sequences of MNIST images are not naturally sequential inputs i.e., there are no sequential dynamics underlying them. We thus examine the models on the MovingMNIST dataset [58]. Each video in this dataset consists of 20 frames of 2 MNIST digits moving inside a $64 \times 64$ patch. Due to the fixed sequence length, for experiments with MovingMNIST we vary the total number of sequences to memorize and fix the sequence length to the first 10 frames of the videos. The performance of the models is shown in Fig. 3B. On average, the recall MSE of MCAHN has a slower increase than that of the single-layer tPC as the total number of sequences increases. However, the performance of MCAHN has very large variations across all sequence numbers. To probe into this observation, we visually examined 3 examples of the MovingMNIST movies recalled by MCAHN and our single-layer tPC in Fig. 4A, when the total number of sequences to memorize is 40. MCAHN produces very sharp recalls for the first 2 example sequences, but totally fails to recall the third one by converging to a different memory sequence after the red triangle in Fig. 4A. On the other hand, the recall by single-layer tPC is less

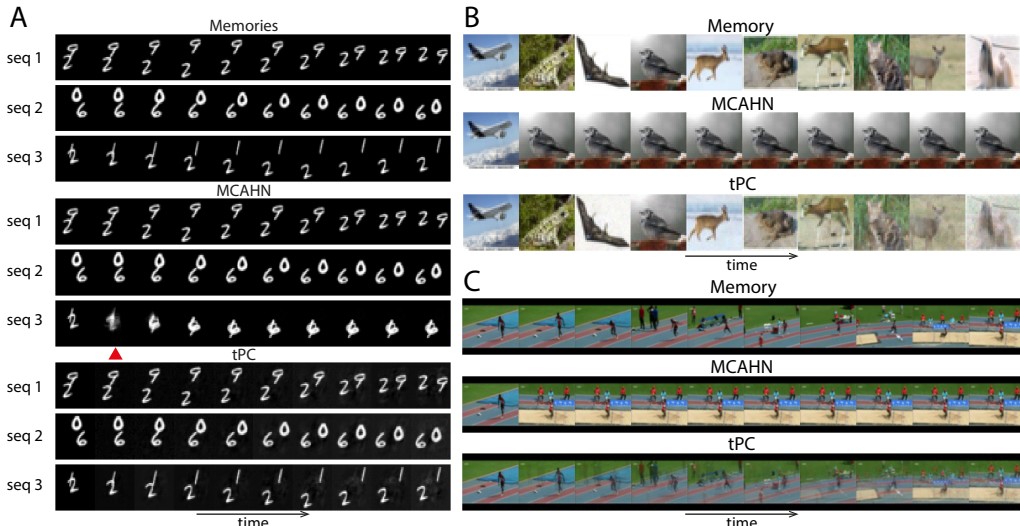

Figure 4: Visual results of offline memory recall with 3 datasets. A: MovingMNIST. B: CIFAR10. C: UCF101.

sharp but stably produces the correct sequence. This phenomenon can be understood using the UHN framework [7]: in Eq. 5, when $\beta$ is large, the softmax separation function used by MCAHN will assign a weight close to $1$ to the memory whose preceding frame is most similar to $\mathbf{q}$ (measure by dot product), and weights close to $0$ to all other memories, which results in the sharp recall. In contrast, our single-layer tPC model uses an identity separation function that fails to suppress the incorrect memories, resulting in blurry recalls. Importantly, however, the failure of MCAHN in sequence 3 in Fig. 4A suggests that there are "strong attractors" in the memories with an undesirable advantage in dot product similarity which resulted in the large variance in the numerical results, and is addressed by the whitening step in single-layer tPC as it effectively normalized the patterns before dot product.

The importance of whitening in tPC is further demonstrated in our experiments with random sequences of CIFAR10 [59] images and movies from the UCF101 [60] dataset, shown in Figs. 4B and C. When recalling these colored sequences, MCAHN can easily converge to strong attractors preceded by frames with many large pixel values that lead to large dot products e.g., the third image in the CIFAR10 example with bright backgrounds, and the penultimate frame in the UCF101 example with a large proportion of sand, which give their subsequent frames large similarity scores. This problem is consistent with earlier findings with static memories [8], and is circumvented in our single-layer tPC model with the whitening matrix $\mathbf{M}$ normalizing the pixel values across the sequence, yielding the correct and more stable memory recalls. However, they are less sharp due to the identity separation function. For all the experiments in Fig. 3B and Fig. 4, we used offline recall to make the tasks more challenging and more consistent with reality. Results with online recalls are shown in SM.

## 4.3 Comparison to behavioral data in sequential memory experiments

We further demonstrated the biological relevance of tPC by comparing it to data from behavioral experiments in sequential memory. In Fig 5A, our 2-layer tPC is compared with Crannell and Parrish's [61] study on sequence length's impact on serial recall of English words. Using one-hot vectors to represent letters (i.e., each "word" in our experiment is an ordered combination of one-hot vectors of "letters", a minimal example of a word with 3 letters being: $[0, 1, 0], [1, 0, 0], [0, 0, 1]$), we demonstrate accuracy as the proportion of perfectly recalled sequences across varying lengths. Our model aligns consistently with experimental data in [61] as well as the recurrent network model by Botvinick and Plaut [48], displaying a sigmoidal accuracy drop with increasing sequence length.

Fig 5B introduces a qualitative comparison to Henson's [62] experimental data, examining primacy/recency effects in serial recall of letters. These effects involve higher accuracy in recalling early (primacy) and late (recency) entries in a sequence, with the recency effect slightly weaker than the primacy effect. Using one-hot vectors and a fixed sequence length 7 (6 positions are shown as the first position is given as the cue to initiate recall in our experiment), we visualize recall frequency at different positions across simulated sequences (100 repetitions, multiple seeds for error bars).

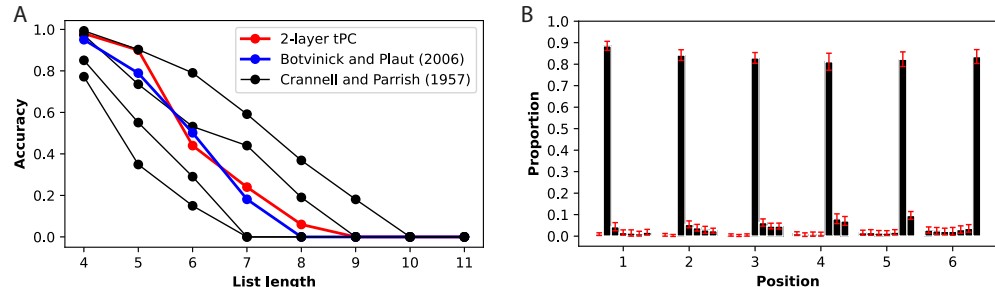

Figure 5: Replicating behavioral data with tPC. A: Experimental data from [61] that studies the impact of sequence length on serial recall of English words and the replications by Botvinick and Plaut [48] and tPC. B: tPC replicates the primacy/recency effects in serial recall experiments [62].

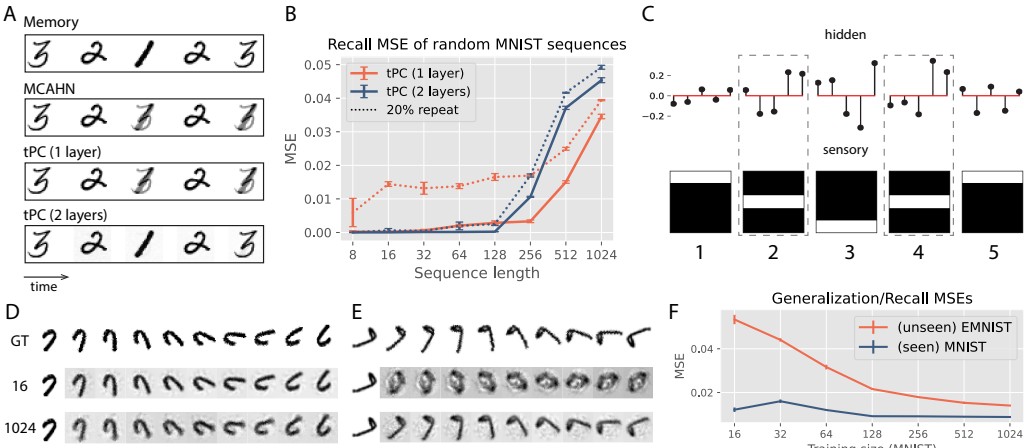

Figure 6: Context representation and generalization of tPC. A: A simple aliased example with MNIST. B: Numerical investigation into the impact of aliased or repeating elements on model performance. C: Different latent representations of aliased inputs by the 2-layer tPC. D/E: Recall/generalization of sequences with rotational dynamics. "GT" stands for ground truth and 16 and 1024 are the numbers of training sequences. F: Recall and generalization MSE of seen and unseen rotating MNIST images. Error bars obtained with 5 seeds.

Each bar in Fig 5B indicates the frequency of an entry at a particular position being recalled at each position. Our 2-layer tPC reproduces primacy/recency effects, albeit weaker than the model in [62] and previous models [48]. Additionally, the model tends to recall neighboring entries upon errors, echoing Henson's data. We attribute the weaker effects to tPC's memory storage in weights, leading to overall improved performance across positions. Details for these experiments are shown in SM.

## 4.4 tPC develops context-dependent representations

In Fig. 3, we plotted the recall MSE of the 2-layer tPC model in MovingMNIST, which is similar to that of the single-layer tPC. The close performance of these models raises the question of whether and when the hidden layer and hierarchical processing of sequential inputs are necessary. Inspired by earlier neuroscience theories that the hippocampus develops neuron populations signaling the sensory inputs and the context of the inputs separately [3, 11], we hypothesize that the hidden neurons in our model represent *when* an element occurs in a memory sequence i.e., its context. We thus designed a sequential memory task with aliased, or repeating inputs at different time-steps [28, 63]. An example of such an aliased sequence can be seen in Fig. 6A, where the second and the fourth frames of a short MNIST sequence are exactly the same ("2"). Recalling such a sequence is inherently more challenging as the models have to determine, when queried with the "2" during recall (either online or offline), whether they are at the second or the fourth step to give the correct recall at the next step ("1" or "3"). As can be seen in Fig. 6A, both MCAHN (which can be regarded as a single-layer network [7]) and single-layer tPC fail in this task, recalling an average frame of "1" and "3" after the

aliased steps, whereas the 2-layer tPC can recall the correct sequence. We then conducted a numerical investigation into this problem. We first plotted the (online) recall MSEs of the single- and 2-layer tPC models in random MNIST sequences (sampled from the training set) of varying lengths, which are shown as the solid lines in Fig. 6B. We then randomly replaced $20\%$ (rounded up to the closest integer) of the elements in these sequences with a single digit from the test set of MNIST so that each sequence now has $20\%$ repeating i.e., aliased elements, and plotted the recall MSEs as the dotted lines in Fig. 6B. The result suggests that sequences with aliased inputs affect the single-layer model much more than it affects the 2-layer one, producing significantly larger MSEs than recalls without aliased inputs. It is worth noting that aliased inputs are ubiquitous in natural sequential memories. In SM, we provide a natural example from the UCF101 dataset where the 2-layer tPC successfully de-aliased repeating inputs whereas the MCAHN and single-layer tPC failed.

To further understand the mechanism underlying the 2-layer tPC with aliased memories, we used a simpler synthetic sequential memory shown in Fig. 6C bottom, where a white bar moves first down and then up in a $5 \times 5$ frame so that the steps 2 and 4 are aliased [28]. We then trained a 2-layer tPC with a hidden size 5 to memorize this sequence and queried it offline. The smaller hidden size allows us to plot, when the recall dynamics (Eq. 11) have converged, the exact hidden activities in Fig. 6C top, where each vertical line represents the activity of a hidden value neuron $\hat{z}^\mu$. As can be seen in the circled time-steps, the 2-layer model represents the aliased inputs differentially in its hidden states, which helps it recall the next frame correctly. This property is consistent with early observations in neuroscience that when memorizing sequences, the hippocampus develops a conjunction of neurons representing individual inputs, as well as neurons signaling the temporal context within which an individual appears [3, 11]. In our simple 2-layer model, the "sensory" layer represents individual inputs, whereas the hidden layer plays the role of indexing time and context.

### 4.5 tPC generalizes learned dynamics to unseen sequences

After memorizing a large number of training sequences sharing underlying dynamics, can tPC generalize the dynamics to unseen sequences? In this experiment, we train the 2-layer tPC with sequences of rotating MNIST digits of identical rotating dynamics (i.e., the digits are rotating towards a fixed direction with a fixed angle at each time step) and vary the number of training sequences ("training size"). An example of these rotating MNIST digits can be seen in Fig 6D, row "ground truth". The model's performance is assessed by its ability to rotate both seen MNIST digits and unseen EMNIST letters. For small training sizes (16), tPC can recall seen rotating digits but struggles with generalizing to unseen letters (Fig 6D and E, second row). Increasing the training size to 1024 improves generalization, evident in clearer rotating sequences (Fig 6D and E, bottom row). Panel F quantitatively confirms this trend: the generalization MSE on unseen EMNIST drops as MNIST training size increases, indicating the model learns the underlying dynamics. Interestingly, the recall MSEs for seen MNIST sequences also decrease due to the model extracting rotational dynamics from the larger training set, differing from the behavior observed in random MNIST sequences (Fig 3A). Generalization and the capability of developing contextual representations that disambiguate aliased inputs are two critical functions underlying the flexible behavior of animals, thus connecting our tPC to models of cognitive maps in the hippocampus and related brain regions [51].

## 5    Conclusion

Inspired by experimental and theoretical discoveries in neuroscience, in this work we have proposed a temporal predictive coding model for sequential memory tasks. We have shown that our tPC model can memorize and recall sequential inputs such as natural movies, and performs more stably than earlier models based on Asymmetric Hopfield Networks. We have also provided a theoretical understanding of the stable performance of the tPC model, showing that it is achieved by an additional statistical whitening operation that is missing in AHNs. Importantly, this whitening step is achieved implicitly by a plausible neural circuit performing local error minimization. Moreover, our 2-layer tPC has exhibited representational and behavioral properties consistent with biological observations, including contextual representations and generalization. Overall, our model has not only provided a possible neural mechanism underlying sequential memory in the brain but also suggested a close relationship between PC and HN, two influential computational models of biological memories. Future directions include systematic investigations into tPC with more than 2 layers and modelling cognitive maps with tPC.

## 6    Acknowledgement

This work has been supported by Medical Research Council UK grant MC_UU_00003/1 to RB, an E.P. Abraham Scholarship in the Chemical, Biological/Life and Medical Sciences to MT, and UKRI Future Leaders Fellowship to HB (MR/W008939/1). The authors would like to acknowledge the use of the University of Oxford Advanced Research Computing (ARC) facility in carrying out this work. http://dx.doi.org/10.5281/zenodo.22558

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
