# Sequential Memory with Temporal Predictive Coding
# Supplementary Materials

**Mufeng Tang, Helen Barron, Rafal Bogacz**
MRC Brain Network Dynamics Unit, University of Oxford, UK
{mufeng.tang, helen.barron, rafal.bogacz}@bndu.ox.ac.uk

## 1 Algorithms

In Algorithm 1 we present the memorizing and recalling procedures of the single-layer tPC.

---
**Algorithm 1** Memorizing and recalling with single-layer tPC

---

1: ▷ *Training/memorization*
2: **while W** not converged **do**
3:     **for** $\mu = 2, ..., P$ **do**
4:         Input: $\mathbf{x}^\mu, \mathbf{x}^{\mu-1}$
5:         Update **W**
6:     **end for**
7: **end while**
8:

9: ▷ *Cued recall*
10: **for** $\mu = 2, ..., P$ **do**
11:     Input: $\mathbf{x}^{\mu-1}$ (online) or $\hat{\mathbf{x}}^{\mu-1}$ (offline)
12:     **while** $\hat{\mathbf{x}}^\mu$ not converged **do**
13:         Infer $\hat{\mathbf{x}}^\mu$
14:     **end while**
15:     $R^{tPC}(\mathbf{q}) \leftarrow \hat{\mathbf{x}}^\mu$
16: **end for**

---

In Algorithm 2 we present the memorizing and recalling procedures of the 2-layer tPC.

---
**Algorithm 2** Memorizing and recalling with 2-layer tPC

---

1: ▷ *Training/memorization*
2: **while** $\mathbf{W}_H, \mathbf{W}_F$ not converged **do**
3:     randomly initialize $\hat{\mathbf{z}}^0$
4:     **for** $\mu = 1, ..., P$ **do**
5:         Input: $\mathbf{x}^\mu, \hat{\mathbf{z}}^{\mu-1}$
6:         **while** $\mathbf{z}^\mu$ not converged **do**
7:             Infer $\mathbf{z}^\mu$
8:         **end while**
9:         Update $\mathbf{W}_H, \mathbf{W}_F$
10:         $\hat{\mathbf{z}}^\mu \leftarrow \mathbf{z}^\mu$
11:     **end for**
12: **end while**
13: ▷ *Cued recall*
14: randomly initialize $\hat{\mathbf{z}}^0$
15: **for** $\mu = 1, ..., P$ **do**
16:     Input: $\hat{\mathbf{z}}^{\mu-1}$
17:     **while** $\mathbf{z}^\mu$ and $\hat{\mathbf{x}}^\mu$ not converged **do**
18:         Infer $\mathbf{z}^\mu, \hat{\mathbf{x}}^\mu$
19:     **end while**
20:     $\hat{\mathbf{z}}^\mu \leftarrow \mathbf{z}^\mu$
21:     $R^{tPC}(\mathbf{q}) \leftarrow \hat{\mathbf{x}}^\mu$
22: **end for**

---

It is worth noting that, although in both algorithms we used iterative inference (line 14-16 in Algorithm 1 and line 17-19 in Algorithm 2), these inferential dynamics can be replaced by forward passes in simulation. For the single-layer model the retrieval $R^{tPC}(\mathbf{q})$ can be directly obtained by $R^{tPC}(\mathbf{q}) = \mathbf{W}f(\mathbf{q})$ with the learned **W**, while for the 2-layer model the retrieval $\mathbf{x}^{\mu-1}$ can be obtained by first forward passing the latent $\hat{\mathbf{z}}^\mu = \mathbf{W}_H f(\hat{\mathbf{z}}^{\mu-1})$ and then set the retrieval as $R^{tPC}(\mathbf{q}) = \mathbf{W}_F f(\hat{\mathbf{z}}^\mu)$. Effectively, setting the retrieval directly by forward passes will result in the same retrieval as performing the inferential iterations as they are the fixed points of the inferential

37th Conference on Neural Information Processing Systems (NeurIPS 2023).

dynamics. However, obtaining the retrievals via iterative methods allows us to implement the computations in the plausible neural circuits in Fig. 1 whereas forward passes cannot. Code will be available upon acceptance.

## 2 Proof of Property 1

Here we present the proof for Property 1 in the main text, that the single-layer tPC can be viewed as a "whitened" version of the AHN. Without loss of generality, assume a sequence of zero-mean, real-valued patterns $\mathbf{x}^\mu$, $\mu = 1, ..., P + 1$ is given to the model to memorize. The step-wise objective with an identity non-linearity $f(\cdot)$ for single-layer tPC is:

$$\mathcal{F}_\mu(\mathbf{W}) = \|\mathbf{x}^\mu - \mathbf{W}\mathbf{x}^{\mu-1}\|_2^2 \tag{1}$$

The weight $\mathbf{W}$ is then updated at each time-step once, and the whole sequence is presented for multiple iterations until $\mathbf{W}$ converges. We now consider the case where we presented the sequence at once i.e., the model now minimizes the following objective at each learning iteration:

$$\mathcal{F} = \sum_{\mu=1}^{P} \|\mathbf{x}^{\mu+1} - \mathbf{W}\mathbf{x}^\mu\|_2^2 \tag{2}$$

which can be viewed as a "batched" version of Eq. 1. Since the objective is now convex (with identity $f(\cdot)$), the fixed point obtained by these two objectives will be the same. The gradient descent update of $\mathbf{W}$ on $\mathcal{F}$ is then:

$$\Delta\mathbf{W} = -\frac{\partial\mathcal{F}}{\partial\mathbf{W}} = \sum_{\mu=1}^{P} \mathbf{x}^{\mu+1}(\mathbf{x}^\mu)^\top - \mathbf{W}\sum_{\mu=1}^{P} \mathbf{x}^\mu(\mathbf{x}^\mu)^\top \tag{3}$$

By setting $\Delta\mathbf{W}$ to 0, we could obtain the optimal $\mathbf{W}$, which we call $\mathbf{W}^{tPC}$:

$$\mathbf{W}^{tPC} = \left(\sum_{\mu=1}^{P} \mathbf{x}^{\mu+1}(\mathbf{x}^\mu)^\top\right)\left(\sum_{\mu=1}^{P} \mathbf{x}^\mu(\mathbf{x}^\mu)^\top\right)^{-1} = \sum_{\mu=1}^{P} \mathbf{x}^{\mu+1}(\mathbf{x}^\mu)^\top \left(\mathbf{X}^\top\mathbf{X}\right)^{-1} \tag{4}$$

where we define $\mathbf{X} = \left[\mathbf{x}^1, ..., \mathbf{x}^P\right]^\top$, the $P \times N$ data matrix. Recall that when presented with a query $\mathbf{q}$, the single-layer tPC update its value nodes to minimize:

$$\mathcal{F}_\mu(\hat{\mathbf{x}}^\mu) = \|\hat{\mathbf{x}}^\mu - \mathbf{W}^{tPC}\mathbf{q}\|_2^2 \tag{5}$$

which will converge to $R^{tPC}(\mathbf{q}) = \mathbf{W}^{tPC}\mathbf{q}$ due to convexity. We can now substitute $\mathbf{W}^{tPC}$ with the expression from Eq. 4 to obtain the retrieval:

$$R^{tPC}(\mathbf{q}) = \mathbf{W}^{tPC}\mathbf{q} = \sum_{\mu=1}^{P} \mathbf{x}^{\mu+1}(\mathbf{x}^\mu)^\top \left(\mathbf{X}^\top\mathbf{X}\right)^{-1}\mathbf{q} \tag{6}$$

It can be immediately seen that the retrieval function of tPC is a special case of the UHN framework, where the similarity function is defined as $\text{sim}(\mathbf{x}^\mu, \mathbf{q}) = (\mathbf{x}^\mu)^\top \left(\mathbf{X}^\top\mathbf{X}\right)^{-1}\mathbf{q}$ and the separation function is identity. Notice that since we assumed a zero-mean sequence (sequences with non-zero mean can be accounted for with a bias term in the objective function Eq. 2), the term $\mathbf{X}^\top\mathbf{X}$ is exactly the covariance matrix of the sequence. Defining it as $\mathbf{\Sigma}$, the retrieval can be written as:

$$R^{tPC}(\mathbf{q}) = \mathbf{W}^{tPC}\mathbf{q} = \sum_{\mu=1}^{P} \mathbf{x}^{\mu+1}(\mathbf{x}^\mu)^\top\mathbf{\Sigma}^{-1}\mathbf{q} \tag{7}$$

Assume a positive definite covariance $\mathbf{\Sigma}$, it is possible to decompose $\mathbf{\Sigma}^{-1}$ as follows:

$$\mathbf{\Sigma}^{-1} = \mathbf{M}^\top\mathbf{M} \tag{8}$$

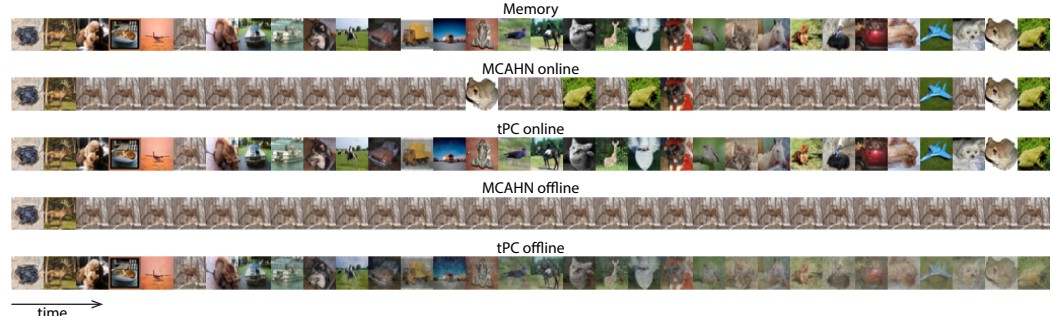

Figure 1: Visual results with CIFAR10 sequences.

The matrix $\mathbf{M}$ is called the whitening matrix, which does not hold a unique value e.g., $\mathbf{M} = \mathbf{\Sigma}^{-\frac{1}{2}}$ or $\mathbf{M} = \mathbf{L}^\top$ where $\mathbf{L}$ is the Cholesky decomposition of $\mathbf{\Sigma}^{-1}$ [1]. Here, we are agnostic about its exact value. When applied to the data sequence, it whitens the data such that (i.e., Eq.16 in the main text):

$$\langle \mathbf{M}\mathbf{x}^\mu (\mathbf{M}\mathbf{x}^\mu)^\top \rangle_\mu = \mathbf{I}_N \tag{9}$$

Therefore, the retrieval of our single-layer tPC with an identity $f(\cdot)$ can be written as:

$$R^{tPC}(\mathbf{q}) = \sum_{\mu=1}^{P} \mathbf{x}^{\mu+1} (\mathbf{M}\mathbf{x}^\mu)^\top \mathbf{M}\mathbf{q} \tag{10}$$

by decomposing $\mathbf{\Sigma}^{-1}$ in Eq. 7 into $\mathbf{M}^\top \mathbf{M}$, which is Eq.15 in the main text and concludes the proof.

## 3 Generation of binary patterns

For the experimental results in Fig. 2A and B, the correlated binary patterns are generated following the approach mentioned in [2]. For a particular pattern dimension $N$, a template $\mathbf{x}^{temp} \in \{-1, 1\}^N$ is first generated. Then, for each of the $\mu = 1, ..., P$ patterns, the $i$th entry of $\mathbf{x}^\mu$ is equal to the $i$th entry of $\mathbf{x}^{temp}$ with probability $0.5 + 0.5b$ where $b$ is the parameter controlling bias. We also invert the sign of each pattern by chance to keep the level of activity constant for each neuron. This whole process is then repeated for multiple trials to average out randomness. The capacity $P_{max}$ is calculated as the maximum $P$ such that the percentage of erroneous entries across these trials is below $0.01$. It can be shown that the correlation between two features of the generated patterns $r(\mathbf{x}_i^\mu, \mathbf{x}_j^\mu)$ is $-b^2$ or $b^2$, by using the identity $r(\mathbf{x}_i^\mu, \mathbf{x}_j^\mu) = \langle \mathbf{x}_i^\mu \mathbf{x}_j^\mu \rangle_\mu - \langle \mathbf{x}_i^\mu \rangle_\mu \langle \mathbf{x}_j^\mu \rangle_\mu$.

## 4 Additional results with CIFAR10 and MovingMNIST

In Fig. 1 we present additional visual results when MCAHN and single-layer tPC are trained to memorize a random CIFAR10 sequence of 32 images and are queried both online and offline during recall. It can be seen that MCAHN, like what we have shown in the main text, recalls memories preceded with images with large pixel values (images are presented to the models as $32 \times 32 \times 3 = 3072$-dimensional vectors where 3 represents the RGB channels) in both online and offline recall regimes, whereas tPC does not suffer from this problem because of the whitening procedure. However, it can be seen that the recall by tPC will gradually become more blurry and noisier when queried offline because the recall errors will accumulate temporally.

These observations are consistent with our numerical results shown in Fig. 3. In Fig. 3A we show the online recall MSE of CIFAR10 sequences by single-layer tPC (with linear $f(\cdot)$) and MCAHN. MCAHN has a much larger MSE than that of the tPC because of the entirely wrong recalls. In offline recalls in Fig. 3B however, tPC will have exploding MSE as soon as $P$ reaches 64 because of the accumulating recall errors. It is worth mentioning that for these experiments, we used a $tanh$ nonlinearity, as the recall error will accumulate to infinity with an identity $f(\cdot)$. This is the only case where identity and $tanh$ are different in our experiments.

In Fig. 2 we also present the online recall results of the models in MovingMNIST, CIFAR10 and UCF101. The results with CIFAR10 are consistent with the discussions above, and the results

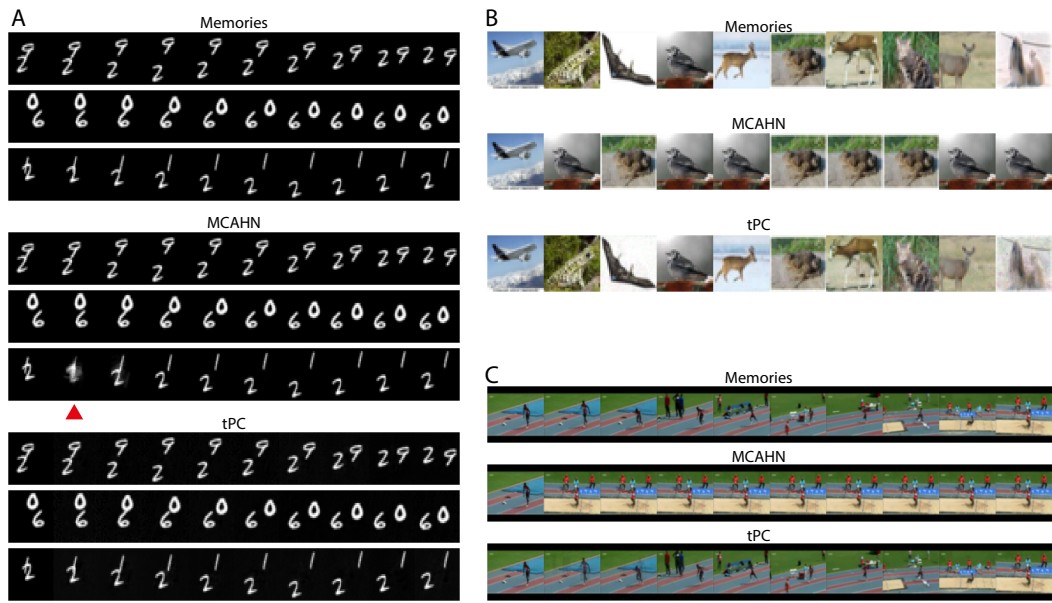

Figure 2: Online recalls with A: MovingMNIST; B: CIFAR10; C: UCF101.

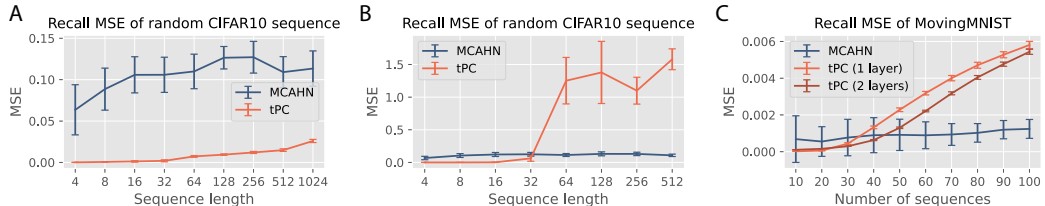

Figure 3: Numerical results with CIFAR10 and MovingMNIST. A: *Online* recall MSE of random CIFAR10 sequences; B: *Offline* recall MSE of random CIFAR10 sequences, with a $tanh$ nonlinearity; C: *Offline* recall MSE of movingMNIST dataset, with a $tanh$ nonlinearity.

with UCF101 is clearer with online queries than those with offline queries. Moreover, although in MovingMNIST MCAHN still suffers from the wrong attractor problem (red triangle in Fig. 2A), the online query can prevent it from staying in the wrong attractor. This is consistent with our numerical observation in Fig. 3C, where the performance of MCAHN in online recall of MovingMNIST is better than that of the tPC models.

# 5    A natural example of aliased sequences from UCF101

In Fig 4 we show a natural example of aliased sequences where a movie of a human doing push-ups is memorized and recalled by the model. The frames at the second and the fifth steps are almost identical, leading to inaccurate predictions of the single-layer models (including MCAHN and single-layer tPC) at the next time steps. On the other hand, the 2-layer tPC performs well and produces sharp and correct recalls.

# 6    Implementation details of tPC

The following table provides the hyperparameters used in our experiments and their corresponding figures. Note that for the 2-layer tPC, we used fixed inference step size 1e-2 and inference steps 100 for Eqs.11 and 14 in the main text, as we did not find any significant impact of these variables on the results. All computations were performed on a single Tesla V100 GPU. Code is available at: https://github.com/C16Mftang/sequential-memory.

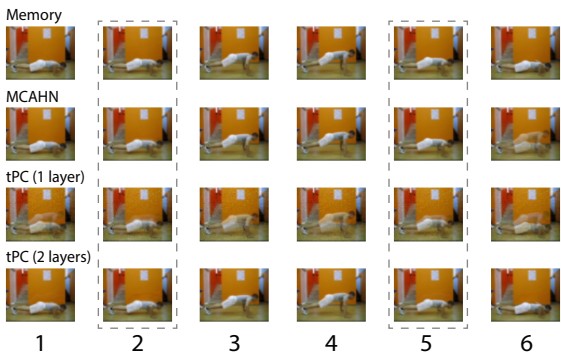

Figure 4: A natural aliased example from the UCF101 dataset, showing a human doing push-ups.

| Data | Figures | Model | Input size | Latent size | Learning rate | Learning epochs |
|------|---------|-------|-----------|-------------|---------------|-----------------|
| Binary | 2A&B,5A&B | 1-layer | varying | N/A | 5e-1 | 800 |
| MNIST | 2C,3A | 1-layer | 784 | N/A | 1e-4 | 800 |
| MNIST | 6A&B | 2-layer | 784 | 480 | 1e-4 | 800 |
| MovingMNIST | 3B,4A | 1-layer | 1024 | N/A | 2e-4 | 800 |
| MovingMNIST | 3B | 2-layer | 1024 | 630 | 2e-4 | 800 |
| CIFAR10 | 4B | 1-layer | 3072 | N/A | 2e-5 | 1000 |
| UCF101 | 4C | 1-layer | 12288 | N/A | 1e-5 | 1000 |
| UCF101 | 4C | 2-layer | 12288 | 7600 | 1e-5 | 1000 |
| One-hot "letters" | 5 | 2-layer | varying | 6 | 5e-2 | 300 |
| rotatingMNIST | 6D&E&F | 2-layer | 784 | 480 | 1e-4 | 800 |

Table 1: Hyperparameters when training tPC models