# OpenReview forum: "Sequential Memory with Temporal Predictive Coding"
_NeurIPS.cc/2023/Conference — NeurIPS 2023 poster_

### Official Review · Reviewer_zZJv · 2023-06-26

**Soundness:** 2 fair
**Presentation:** 2 fair
**Contribution:** 1 poor
**Rating:** 3
**Confidence:** 4

**Summary:**

This paper proposes to use Predictive Coding Networks for temporal association of sequences.

**Strengths:**

The paper is well-structured and easy to follow. The motivation is clear: a deep network model with biologically plausible learning algorithms for sequence learning.

**Weaknesses:**

1. This paper lacks of novelty and is in fact a trivial extension of [1]. In [1], the single-layer and two-layer predictive coding networks are proposed to associate the input and the output. In this paper, the input is replaced with frame x[t] in a sequence and the output is replaced with frame x[t+1].

2. Property 1 in Section 4 is a trivial result for linear regression. It should be noted that Property 1 is not rigorously presented. A condition that the data covariance matrix must be full-ranked should be imposed.

3. Lack of robust retrieval evaluation. In [1] and classic Hopfield networks, the model can recover the stored memories given noisy initialize state. Is the model in this paper robust to noise for sequence storage? Further experimental evaluation is needed.

4. Missing references [2,3]. [2] is the very first work for temporal sequence association. How do the authors compare their work to [3], which is also about predictive coding for sequences?

[1] Associative Memories via Predictive Coding. Tommaso Salvatori, Yuhang Song, Yujian Hong, Simon Frieder, Lei Sha, Zhenghua Xu, Rafal Bogacz, Thomas Lukasiewicz. arXiv, 2021.
[2] Learning Patterns and Pattern Sequences by Self-Organizing Nets of Threshold Elements. S.-I. Amari. IEEE Transactions on Computers, 1972.
[3] Deep Predictive Coding Networks for Video Prediction and Unsupervised Learning. William Lotter, Gabriel Kreiman, David Cox. ICLR, 2017.


**Questions:**

See above.

**Limitations:**

See above.

---

> ### Author Rebuttal · Authors · 2023-08-08
>
> We thank the reviewer for their constructive comments on our paper. Specific responses are provided below and we kindly request that the reviewer consider reevaluating their score in light of our responses:
> ***
> > “This paper lacks of novelty and is in fact a trivial extension of [1]. In [1], the single-layer and two-layer predictive coding networks are proposed to associate the input and the output. In this paper, the input is replaced with frame x[t] in a sequence and the output is replaced with frame x[t+1]”
>
> We agree with the reviewer that this work extends [1] and this is also what we claimed in Related Works. However, the extension is not trivial as [1] only addresses static memories whereas our model performs sequential memory, arguably a more important type of memory, a point we made at the beginning of the paper. Notably, the seminal introduction of Asymmetric Hopfield Networks by Sompolinsky and Kanter (1986) can be viewed as an analogous extension of the original Hopfield Network towards sequential memory. Such advancements, aiming to address broader biological phenomena, should not be dismissed as trivial. Therefore, we wouldn’t consider our work as trivially extending [1].
>
> The models in [1] did not directly associate input and output; they learned configurations representing inputs and recalled them when given a corrupted cue. Moreover, they didn't propose single-layer models. Our work differs by introducing temporal predictions for sequential inputs, unlike [1], where predictions were limited to interactions between layers. Thus, our model's innovation extends beyond replacing [1]'s input and output.
> ***
> > “Property 1 in Section 4 is a trivial result for linear regression. It should be noted that Property 1 is not rigorously presented. A condition that the data covariance matrix must be full-ranked should be imposed.”
>
> We thank the reviewer for pointing out that a full-rank covariance matrix condition is needed for Property 1 to hold. We will add it to the paper.
>
> However, we respectfully disagree with the reviewer that Property 1 is a trivial result from linear regression. While it does stem from regressing x[t+1] to x[t] within our model, its significance is far from trivial for the following reasons:
>
> 1. Property 1 originates from a biologically plausible neural circuit illustrated in Figure 1 of our original paper. This depiction elucidates how the seemingly straightforward linear regression can be executed in a biologically plausible manner within the hippocampus, with local computations and Hebbian plasticity. Consequently, the result holds substantial importance from a biological context.
>
> 2. The expression of Eq 15 in Property 1 is in a form akin to Universal Hopfield Network (Millidge et al., 2022), which establishes a vital connection between two influential computational models: predictive coding and Hopfield Network. This linkage between independently proposed models offers a significant contribution to the realm of computational modeling of neural systems.
>
> 3. Property 1 offers an insightful interpretation of linear regression coefficients. By decomposing $(X^TX)^{-1}$, it elucidates how linear regression employs a whitened similarity function to compare the input with training data, leveraging this similarity score to weigh target variables and compute the weighted sum of all targets. This interpretation has a noteworthy contribution to the theoretical comprehension of linear regression.
> ***
> > “Lack of robust retrieval evaluation. In [1] and classic Hopfield networks, the model can recover the stored memories given noisy initialize state. Is the model in this paper robust to noise for sequence storage?”
>
> We acknowledge the reviewer's interest in noisy query retrieval and have incorporated experimental outcomes in the attached PDF file in global responses (Fig2). We conducted experiments similar to Fig3A in our original paper, introducing Gaussian noise with varying standard deviation (std) to the recall query. Fig2A illustrates single-layer tPC's performance in retrieving from noisy queries with std 0.1 (top row) and 1.0 (bottom row). As noise levels increase, recall quality diminishes, yet tPC's recall inference progressively refines and denoises retrieved images. In Fig2B, the same evaluation is presented for MCAHN, showcasing consistent retrieval of clear and sharp images. This clarity stems from MCAHN's softmax separation function (Eq 5), absent in tPC (Eq 15). Fig2C provides a quantitative assessment of noise impact on recall MSE for memorized sequence lengths of 16 and 32. Generally, both tPC and MCAHN experience increased recall MSE with higher noise levels. However, while MCAHN performs better than tPC at sequence length 16, the reverse holds true at length 32. This observation, combined with Fig3A in our original paper, affirms sequence length's predominant role in recall MSE under these noise levels. It's noteworthy that MCAHN's recall MSE variance is substantial due to the softmax separation function's potential for convergence to erroneous sequence entries, as explained in our original paper.
> ***
> > “Missing references [2,3]. [2] is the very first work for temporal sequence association. How do the authors compare their work to [3], which is also about predictive coding for sequences”
>
> We have added [2] to our literature review, in global responses. The “predictive coding” used in [3] is in a broader sense: they trained a deep network using backprop, end-to-end, whereas our model is trained using local and Hebbian learning rules derived from local prediction errors. However, we agree that it would be an interesting future direction to investigate whether tPC could achieve comparable results to [3]. We will add this to our Discussion and Related Works section.
> ***
> > References
> - Sompolinsky, Haim, and Ido Kanter. Physical review letters. 1986.
> - Millidge, Beren, et al. International Conference on Machine Learning. 2022.

---

> > ### Comment · Reviewer_zZJv · 2023-08-16
> >
> > I have read the rebuttal and retain my original rating.

---

### Official Review · Reviewer_SUeD · 2023-07-03

**Soundness:** 3 good
**Presentation:** 3 good
**Contribution:** 3 good
**Rating:** 7
**Confidence:** 4

**Summary:**

The paper presents work on (relatively) biologically-plausible neural networks for remembering sequences of inputs, extending work on temporal predictive coding nets (a simple architecture of a layer of neurons for feature values and a layer for prediction error, with some interneurons) and asymmetric modern hopfield networks.
Analysis shows a direct link between temporal Predictive Coding networks and Asymmetric Hopfield Networks, with improved performance vs AHN for correlated patterns reflecting the implicit whitening process built into the tPC net.
The multilayer tPC net develops interesting representations of items and context while solving the problem of storing sequences with repeated items.

**Strengths:**

There are nice clear theoretical results to explain the links between tPC and AHN, and new results showing good performance in challenging sequential memory tasks with complex repeated images.
The development of interesting higher order representations of sequential order in this simple-to-analyse system will be of interest to neuroscientists given the development of these representations in the mammalian brain.

**Weaknesses:**

The sequential memory solutions considered here use changes to connection weights to store the sequence, these might be compared with deep networks that are able to reproduce a sequence having been pretrained on similar sequences, but not the one in question (the first example I think is Botvinick & Plaut, Psych Rev, 2006; now transformers).
It seems that the only advantage in performance, compared to the AHN, is the whitening process (which could easily be added to AHN), but perhaps this really reflects the fact that this is a biologically plausible implementation of AHN (which introduces whitening as a by product) - perhaps it could be compared to alternative ways of implementing AHN in a biologically plausible way (if they exist)?

**Questions:**

Would an alternative be to explicitly make predictions or learn associations between items that are more than one place apart in the list?
How does this model compare to those explicitly involving a sequential contextual signal (e.g. the "temporal context model")?

**Limitations:**

See above for potential limitations/comparisons that could be discussed

---

> ### Author Rebuttal · Authors · 2023-08-08
>
> We thank the reviewer for their constructive comments on the additional references and possible extensions of our model. Specific responses are given below:
> ***
> > “The sequential memory solutions considered here use changes to connection weights to store the sequence, these might be compared with deep networks that are able to reproduce a sequence having been pretrained on similar sequences, but not the one in question (the first example I think is Botvinick & Plaut, Psych Rev, 2006; now transformers)”
>
> We examined the generalization capability of our 2-layer tPC in an experiment shown in Fig1 of the attached PDF file in global responses. Details of these experiments and results can be found in global response, “Description of Figure 1”.
>
> However, we did not compare with Botvinick and Plaut (2006) directly as their sequential memory task is different from ours: their model was trained to memorize **and** recall sequences after a given recall cue. The testing phase is performed on unseen data, to examine whether the model can memorize and recall the new unseen sequences. On the other hand, our model is trained only to memorize, and recall is triggered by presenting to the model the first item in one of the training sequences. This difference in task prevents us from comparing with their model directly.
>
> We value the reviewer's insight regarding Botvinick and Plaut (2006), which prompted us to incorporate a comparison with experimental data. The results are shown in Fig3 of the attached file to global responses and descriptions of the results can be found in the global response, “Description of Figure 3”.
> ***
> > “perhaps it could be compared to alternative ways of implementing AHN in a biologically plausible way (if they exist)”
>
> Currently, we are unaware of any other biologically plausible implementation of AHN that seamlessly incorporates the whitening operation. A possible approach is to combine the implementation in Chaudhry et al. (2023) with circuits performing statistical whitening such as in Duong et al. (2023). However, this requires further thinking and experiments and we defer it for future explorations.
> ***
> > “Would an alternative be to explicitly make predictions or learn associations between items that are more than one place apart in the list? “
>
> We agree this is an interesting direction to follow, as it may lead to a richer representation of contexts and time in the latent layers. However, this is beyond the scope of the current paper where we aim to explore the computational principle of the single-layer tPC and present preliminary results of the 2-layer tPC. We aim to explore this question in future works.
> ***
> > “How does this model compare to those explicitly involving a sequential contextual signal (e.g. the "temporal context model")?”
>
> Similar to the previous point, we aim to explore the contextual signal in the 2-layer tPC in future works. However, we appreciate the additional reference on Temporal Context Model and have added it to our literature review in the global responses.
> ***
> > References:
> - Duong, Lyndon R., et al. "Statistical whitening of neural populations with gain-modulating interneurons." arXiv preprint arXiv:2301.11955 (2023).
> - Chaudhry, Hamza Tahir, et al. "Long Sequence Hopfield Memory." arXiv preprint arXiv:2306.04532 (2023).
> - Matthew M Botvinick and David C Plaut. Short-term memory for serial order: a recurrent neural network model. Psychological review, 113(2):201, 2006.

---

> > ### Comment · Reviewer_SUeD · 2023-08-17
> >
> > Thank you for the specific response to my comments, and the interesting global response. I think the new simulations strengthen the paper and reinforce my choice of a score of 7. I will increase my confidence in this score.

---

### Official Review · Reviewer_EgZ5 · 2023-07-06

**Soundness:** 3 good
**Presentation:** 3 good
**Contribution:** 2 fair
**Rating:** 6
**Confidence:** 3

**Summary:**

The authors propose a temporal predictive coding model that can memorize and recall sequences. The model performs better than a model based on asymmetric Hopfield networks. The authors provide a theoretical evaluation end explain the reasons for better performance. This work is inspired by neuroscience results and the authors argue that it establishes a possible computational mechanism underlying sequential memory in the brain.

**Strengths:**

This paper proposes a new model for learning sequences using temporal predictive coding. The method is well explained, and results consist of several experiments showing better performance than when using an asymmetric Hopfield network. The authors provide a connection between temporal predictive coding and asymmetric Hopfield network - they identified how temporal predictive coding actually performs the same operation as asymmetric Hopfield network but with an implicit statistical whitening step during memory recall. They showed that when using multi-layer temporal predictive coding, the model develops latent representations of contextual information in sequential memories.

**Weaknesses:**

The authors mentioned that this work helps establish a possible computational mechanism underlying sequential memory in the brain. In its current form, the paper lacks direct comparison with neural or behavioral data. Behavioral tasks such as free recall could be used to evaluate if the properties of the sequential memory resemble those in the brain.

**Questions:**

While the authors mentioned some of the related work, here are a few more papers that build neural models of memory for sequences.

Graves et al. 2014 Neural Turing machines
Voelker et al. 2019 Legendre memory units: Continuous-time representation in recurrent neural networks
Eliasmith et al. 2013 A large-scale model of the functioning brain
Howard et al. 2014 A unified mathematical framework for coding time, space, and sequences in the hippocampal region
Whittington et al. 2020 The Tolman-Eichenbaum machine: unifying space and relational memory through generalization in the hippocampal formation

The authors could comment on similarities/differences with respect to these approaches or perform a comparison.

**Limitations:**

The authors provided a sentence about future directions, but not about the limitations.

---

> ### Author Rebuttal · Authors · 2023-08-08
>
> We thank the reviewer for their constructive comments on connecting our models to behavioral data and additional references. Specific responses are provided below and we kindly request that the reviewer consider reevaluating their score in light of our responses:
> ***
> > “In its current form, the paper lacks direct comparison with neural or behavioral data.”
>
> We have added a comparison to behavioral data and the results can be seen in Fig3 of the attached PDF file in global responses. In Fig3A, our 2-layer tPC is compared with Crannell and Parrish's (1957) study on sequence length's impact on serial recall of English letters/words. Using one-hot vectors to represent letters/words (a minimal example with a sequence 3 letters/words can be: [0,1,0], [1,0,0], [0,0,1]), we demonstrate accuracy as the proportion of perfectly recalled sequences across varying lengths. Our model aligns consistently with experimental data in Crannell and Parrish (1957) as well as the model by Botvinick and Plaut (2006), displaying a sigmoidal accuracy drop with increasing sequence length.
>
> Fig3B introduces a qualitative comparison to Henson's (1998) experimental data, examining primacy/recency effects in serial recall. These effects involve higher accuracy in recalling early (primacy) and late (recency) entries in a sequence, with the recency effect slightly weaker than the primacy effect. Using one-hot vectors and fixed sequence length, we visualize recall frequency at different positions across simulated sequences (100 repetitions, multiple seeds for error bars). Each bar in Fig3B indicates the frequency of an entry at a particular position being recalled at each position. Our 2-layer tPC reproduces primacy/recency effects, albeit weaker than Henson (1998) and previous models (Botvinick and Plaut, 2006). Additionally, the model tends to recall neighboring entries upon errors, echoing Henson's data. We attribute the weaker effects to tPC's memory storage in weights, leading to overall improved performance across positions.
> ***
> > “While the authors mentioned some of the related work, here are a few more papers that build neural models of memory for sequences.”
>
> We have added a paragraph to the global responses, where we reviewed the papers that the reviewer pointed out. This paragraph will be added to the Related Works section in the camera-ready version of our paper.
> ***
> > "The authors provided a sentence about future directions, but not about the limitations."
>
> This work is limited to modeling sequential memories, whereas the model can possibly be extended to address other functionalities of the hippocampus, such as generalization, based on our initial experiments. We aim to explore this direction further in future works.
> ***
> > References
>
> - Henson, Richard NA. "Short-term memory for serial order: The start-end model." Cognitive psychology 36.2 (1998): 73-137.
> - Crannell, C. W., and J. M. Parrish. "A comparison of immediate memory span for digits, letters, and words." The Journal of Psychology 44.2 (1957): 319-327.
> - Matthew M Botvinick and David C Plaut. Short-term memory for serial order: a recurrent neural network model. Psychological review, 113(2):201, 2006.

---

> > ### Comment · Reviewer_EgZ5 · 2023-08-19
> >
> > Thank you for your detailed response. I believe the additional experiments improved the manuscript and I adjusted my score accordingly.

---

### Official Review · Reviewer_vgoV · 2023-07-07

**Soundness:** 4 excellent
**Presentation:** 3 good
**Contribution:** 3 good
**Rating:** 7
**Confidence:** 4

**Summary:**

The paper generalizes predictive coding as a method of training neural networks to Hopfield networks, giving a model of temporal predictive coding (tPC).  tPC proves itself able to memorize discrete sequences at a level competitive with Asymmetric Hopfield Networks in experiments, and provides an intriguing hint as to the potential function of statistical whitening in the hippocampus.

**Strengths:**

The authors provide a strong mathematical grounding and linkage to the predictive coding literature. Temporal predictive coding shows the interesting strength that as the correlation between features in a sequence increases, it does not appear to significantly lose much of its capacity -- no doubt due to the implicit statistical whitening.  They perform an experimental evaluation against sequential versions of the Modern Hopfield Network and the Modern Continuous Hopfield Network, though not against non-Hopfield sequence learning or cognitive mapping models.

**Weaknesses:**

The authors overclaim about biological/neural "memories" from the first sentence of the abstract. This becomes important because lossless sequence memorization is not what the hippocampus does, and if it did, it would be useless.  This leaves the major unaddressed question being: can the paper's tPC model generalize to unseen but reasonably similar sequences?  What features of the sequences can change without disrupting memorization?

**Questions:**

Can the authors address any of the literature in the learning of cognitive maps in the hippocampus?  Those often provide a connection to discrete event sequences and would give the authors a baseline to compare to beyond just the Hopfield networks they use.

**Limitations:**

Are the authors aiming their model and their claims at the computational or algorithmic Marr levels of analysis?  Even if they're aiming at the algorithmic level, predictive coding has been proposed to approximate backpropagation in certain limits, and so they could have compared against backprop-trained memory models in which the backprop training was replaced with a sufficient predictive coding architecture.  Likewise, there are plenty of cognitive mapping models at the computational model which are not analytically tractable, but which do admit sampling/Monte Carlo implementations that can attain neural plausibility.  Why the restricted class of comparisons?  Are the authors specifically proposing that the brain performs all necessary computations in closed form?

EDIT: The authors have fully addressed my concern about closed-form computation, which then places their tPC model within the class of things to which I requested comparison.  This is very good!

---

> ### Author Rebuttal · Authors · 2023-08-08
>
> We thank the reviewer for their constructive comments on generalization and connection of our model to cognitive maps. Specific responses are provided below and we kindly request that the reviewer consider reevaluating their score in light of our responses:
>
> ***
> > “The authors overclaim about biological/neural "memories" from the first sentence of the abstract”
>
> We thank the reviewer for pointing this out. Indeed, the survival of biological agents requires much more beyond sequential memory e.g., generalization. We will change our narrative on the significance of memory in the camera-ready version.
>
> ***
> > “can the paper's tPC model generalize to unseen but reasonably similar sequences? What features of the sequences can change without disrupting memorization?”
>
> We have added experiments to examine the generalization capability of our 2-layer tPC model, and the results are presented in Figure 1 of the PDF file in the global response section. In this experiment, we train the 2-layer tPC with sequences of rotating MNIST digits and vary the number of training sequences (“training size”). An example of these rotating MNIST digits can be seen in Fig 1A, row “ground truth”. The model's performance is assessed by its ability to rotate both seen MNIST digits and unseen EMNIST letters. For small training sizes (16), tPC can recall seen rotating digits but struggles with generalizing to unseen letters (Fig 1A and B). Increasing the training size to 1024 improves generalization, evident in clearer rotating sequences (Fig 1A and B). Panel C quantitatively confirms this trend: the generalization MSE on unseen EMNIST drops as MNIST training size increases, indicating the model learns the underlying dynamics. Interestingly, the recall MSEs for seen MNIST sequences also decrease due to the model extracting rotational dynamics from the larger training set, differing from the behavior observed in random MNIST sequences (Fig 3B in the original paper).
> ***
> > “Can the authors address any of the literature in the learning of cognitive maps in the hippocampus? Those often provide a connection to discrete event sequences and would give the authors a baseline to compare to beyond just the Hopfield networks they use.”
>
> Please see the discussion paragraph in the global response, where we discussed models of cognitive map and many other models related to sequential memory. This paragraph will be added to our Related Works section in the camera-ready version of this paper. In summary, two properties of our model have already shown tPC’s potential connections to cognitive map: 1) generalization 2) latent representation to disambiguate observations (Whittington et al., 2022). However, the focus of this paper is to investigate predictive coding in **sequential memory** whereas cognitive map models concentrate on the hippocampal formation’s role in flexible behavior based on abstracted knowledge. Therefore, we leave the direct comparison between tPC and cognitive map models (like TEM) to future explorations.
> ***
> > “Are the authors aiming their model and their claims at the computational or algorithmic Marr levels of analysis? “
>
> Algorithmic level
> ***
> > “and so they could have compared against backprop-trained memory models in which the backprop training was replaced with a sufficient predictive coding architecture”
>
> It is indeed an interesting direction to explore the relationship between backprop and predictive coding as learning rules in sequential memory, as most discussions on predictive coding as a plausible substitute to backprop focus on supervised learning. However, the focus of this paper is to investigate **whether predictive coding can support sequential memory in the brain and its computational principles** and we do not aim to compare with backprop. We will explore this problem in future works.
> ***
> > “Why the restricted class of comparisons?”
>
> We mainly compare with AHN because of the shared computational principles between tPC and AHN. This is interesting because it connects two classical computational models in neuroscience: predictive coding and Hopfield Nets. However, we do agree with the reviewer that comparing with Hopfield Nets only is restricted. Thus, we added a comparison to a recurrent model by Botvinick and Plaut (2006), shown in Fig3 of the attached PDF file in global responses. Their model has been compared with data collected from benchmark behavioral tasks and thus the comparison to it establishes a connection between tPC and behavioral data. Our result shows that tPC aligns well with the experimental data and can reproduce the effect of sequence length (Fig3A) and the primacy/recency effects (Fig3B).
> ***
> > “Are the authors specifically proposing that the brain performs all necessary computations in closed form?”
>
> No we are not. In fact the computations in tPC are not closed form: Property 1 in our original paper only holds when the temporal prediction is linear. When it is not linear, the retrieval cannot be expressed in closed form and has to be obtained iteratively (See Algorithm2 in SM). We stated this property here to establish a theoretical understanding of tPC and a connection to AHNs.
> ***
> > References:
> - James CR Whittington, Timothy H Muller, Shirley Mark, Guifen Chen, Caswell Barry, Neil
> Burgess, and Timothy EJ Behrens. The tolman-eichenbaum machine: unifying space and
> relational memory through generalization in the hippocampal formation. Cell, 183(5):1249–1263, 2020.
> - James CR Whittington, David McCaffary, Jacob JW Bakermans, and Timothy EJ Behrens. How to build a cognitive map. Nature neuroscience, 25(10):1257–1272, 2022.
> - Matthew M Botvinick and David C Plaut. Short-term memory for serial order: a recurrent neural network model. Psychological review, 113(2):201, 2006.

---

> > ### Comment · Reviewer_vgoV · 2023-08-18
> > **Impressive rebuttal, raising my score**
> >
> > To the authors,
> >
> > Thank you for addressing my concerns about this paper, including the few that were a result of my own confusion.  The global rebuttal and your specific response here indicates, in my view, a significant strengthening of the paper, and I will be raising my score.

---

### Author Rebuttal · Authors · 2023-08-08

**We performed additional experiments as requested by the reviewers and presented the results in the attached PDF file. Since experiments in Fig 1 and 3 are related to the comments from multiple reviewers, we include descriptions of them here for reference:**

> Description of Figure 1:

In this experiment, we train the 2-layer tPC with sequences of rotating MNIST digits and vary the number of training sequences (“training size”). An example of these rotating MNIST digits can be seen in Fig 1A, row “ground truth”. The model's performance is assessed by its ability to rotate both seen MNIST digits and unseen EMNIST letters. For small training sizes (16), tPC can recall seen rotating digits but struggles with generalizing to unseen letters (Fig 1A and B). Increasing the training size to 1024 improves generalization, evident in clearer rotating sequences (Fig 1A and B). Panel C quantitatively confirms this trend: the generalization MSE on unseen EMNIST drops as MNIST training size increases, indicating the model learns the underlying dynamics. Interestingly, the recall MSEs for seen MNIST sequences also decrease due to the model extracting rotational dynamics from the larger training set, differing from the behavior observed in random MNIST sequences (Fig 3B in the original paper).

> Description of Figure 3:

In Fig3A, our 2-layer tPC is compared with Crannell and Parrish's (1957) study on sequence length's impact on serial recall of English letters/words. Using one-hot vectors to represent letters/words (a minimal example with a sequence 3 letters/words can be: [0,1,0], [1,0,0], [0,0,1]), we demonstrate accuracy as the proportion of perfectly recalled sequences across varying lengths. Our model aligns consistently with experimental data as well as the model by Botvinick and Plaut (2006), displaying a sigmoidal accuracy drop with increasing sequence length.

Fig3B introduces a qualitative comparison to Henson's (1998) experimental data, examining primacy/recency effects in serial recall. These effects involve higher accuracy in recalling early (primacy) and late (recency) entries in a sequence, with the recency effect slightly weaker than the primacy effect. Using one-hot vectors and fixed sequence length, we visualize recall frequency at different positions across simulated sequences (100 repetitions, multiple seeds for error bars). Each bar in Fig3B indicates the frequency of an entry at a particular position being recalled at each position. Our 2-layer tPC reproduces primacy/recency effects, albeit weaker than Henson (1998) and previous models (Botvinick and Plaut, 2006). Additionally, the model tends to recall neighboring entries upon errors, echoing Henson's data. We attribute the weaker effects to tPC's memory storage in weights, leading to overall improved performance across positions.

***

**Since all reviewers have pointed us to additional references, we added the following paragraphs of an additional discussion/literature review.**

Beyond Hopfield Networks, many other computational models have been proposed to study the mechanism underlying sequential memory. Theoretical properties of self-organizing networks in sequential memory were discussed as early as in [1]. In theoretical neuroscience, models by Jensen et al. [2] and Mehta et al. [3] suggested that the hippocampus performs sequential memory via neuron firing chains. Other models have suggested the role of contextual representation in sequential memory [4, 5], with contextual representations successfully reproducing the recency and contiguity effects in free recall [6]. Furthermore, Howard et al. [7] proposed that sequential memory is represented in the brain via approximating the inverse Laplacian transform of the current sensory input. However, these models were still at the conceptual level, lacking neural implementations of the computations. Recurrent networks with backpropagation and large spiking neural networks also demonstrate sequential memory [8, 9]. We compare our model with [8] to validate tPC’s alignment with behavior.

Our model is also closely related to the concept of cognitive map in the hippocampal formation
[10 –12], which is often discussed within the context of sequence learning to explain knowledge abstraction and generalization. In this work, we present two preliminary results related to cognitive maps, showing that our tPC model can 1) disambiguate aliased observation via latent representations and 2) generalize with simple sequential dynamics as a result of performing sequential memory [12]. However, as this work centers on memory, we leave cognitive maps for future explorations of tPC.

***

**References:**

[1] S.-I. Amari, IEEE Transactions on computers 100, 1197 (1972).

[2] O. Jensen, M. Idiart, and J. E. Lisman, Learning & Memory 3, 243 (1996).

[3] M. R. Mehta, M. C. Quirk, and M. A. Wilson, Neuron 25, 707 (2000).

[4] G. V. Wallenstein, M. E. Hasselmo, and H. Eichenbaum, Trends in neurosciences 21, 317
(1998).

[5] W. B. Levy, in Psychology of learning and motivation (Elsevier, 1989), vol. 23, pp. 243–305.

[6] M. W. Howard and M. J. Kahana, Journal of mathematical psychology 46, 269 (2002).

[7] M. W. Howard, C. J. MacDonald, Z. Tiganj, K. H. Shankar, Q. Du, M. E. Hasselmo, and H. Eichenbaum, Journal of Neuroscience 34, 4692 (2014).

[8] M. M. Botvinick and D. C. Plaut, Psychological review 113, 201 (2006).

[9] C. Eliasmith, T. C. Stewart, X. Choo, T. Bekolay, T. DeWolf, Y. Tang, and D. Rasmussen, science 338, 1202 (2012).

[10] J. Whittington, T. Muller, S. Mark, C. Barry, and T. Behrens, Advances in neural information processing systems 31 (2018).

[11] J. C. Whittington, T. H. Muller, S. Mark, G. Chen, C. Barry, N. Burgess, and T. E. Behrens, Cell 183, 1249 (2020).

[12] J. C. Whittington, D. McCaffary, J. J. Bakermans, and T. E. Behrens, Nature neuroscience 25, 1257 (2022).

---

### Decision · Program_Chairs · 2023-09-21

**Decision:**

Accept (poster)

**Comment:**

In this theoretical neuroscience study, the authors propose a temporal predictive coding model that can memorize and recall sequences. 3 of the reviewers appreciates the theoretical contributions and comparison to the Asymmetric Hopfield Network.